# Feasibility of an online antigen self-testing strategy for SARS-CoV-2 addressed to health care and education professionals in Catalonia (Spain). The TESTA'T- COVID Project

Cristina Agustí[1,2,3]*, Héctor Martínez-Riveros[1,3,4], Victoria González[1,2,5], Gema Fernández-Rivas[5], Yesika Díaz[1], Marcos Montoro-Fernandez[1], Sergio Moreno-Fornés[1,2], Pol Romano-deGea[1,3], Esteve Muntada[1,3], Beatriz Calvo[6], Jordi Casabona[1,2,3,7]

1 Department of Health, Centre of Epidemiological Studies on Sexually Transmitted Infections and AIDS of Catalunya (CEEISCAT), Generalitat of Catalunya, Badalona, Spain, 2 Spanish Consortium for Research on Epidemiology and Public Health (CIBERESP), Instituto de Salud Carlos III, Madrid, Spain, 3 Germans Trias i Pujol Research Institute (IGTP), Campus Can Ruti, Badalona, Spain, 4 Department of Paediatrics, PhD in Methodology of Biomedical Research and Public Health, Obstetrics and Gynecology and Preventive Medicine, University Autonoma de Barcelona, Badalona, Spain, 5 Clinical Laboratory North Metropolitan Area, Departament of Genetics and Microbiology, Autonomous University of Barcelona Microbiology, Department, Germans Trias i Pujol University Hospital, Badalona, Spain, 6 Occupational Health and Safety Department, Institut Català d'Oncologia (ICO), L'Hospitalet de Llobregat, Spain, 7 Department of Paediatrics, Obstetrics and Gynecology and Preventive Medicine, Universitat Autònoma de Barcelona, Badalona, Spain

* cagusti@iconcologia.net

## Abstract

We aimed to assess the feasibility of TESTA'T COVID strategy among healthcare and education professionals.in Spain during the peak of the 6th wave caused by Omicron variant. Kits were ordered online and sent by mail, participants answered an online acceptability/ usability survey and uploaded the picture of results. 492 participants ordered a test, 304 uploaded the picture (61.8%). Eighteen positive cases were detected (5.9%). 92.2% were satisfied/very satisfied with the intervention; and 92.5% found performing the test easy/very easy. We demonstrated that implementing online COVID-19 self-testing in schools and healthcare settings in Spain is feasible.

## Introduction

It has been estimated that nearly half of the transmissions of SARS-CoV-2 occur from asymptomatic individuals [1]. As for other infections, the screening of asymptomatic individuals at risk of being exposed to SARS-CoV-2 in order to detect and isolate infected persons early is one of the basic non-pharmaceutical preventive interventions shown to decrease incidence at the community level [2]. Antigen-detecting rapid diagnostic tests (Ag-RDTs) have been proposed as suitable tools for point-of-care screening of individuals potentially exposed and have promising performance characteristics for mass population testing [3]. The main advantages of Ag-RDTs include low price, the lack of need for high-tech laboratory referral, and a short

**Data Availability Statement:** All relevant data are within the paper and its Supporting Information files.

**Funding:** This work was supported by the General Direction of the Health Department of the Catalan Government of Catalonia (Spain) [no grant number] and Abbot Laboratories [grant number 8772008]. The funders had no role in study design, data collection and analysis, decision to publish, or preparation of the manuscript.

**Competing interests:** Abbot Laboratories provided the test used in the study. This does not alter our adherence to PLOS ONE policies on sharing data and materials.

turnaround time to provide a result [2,4]. There are concerns about the high rate of false negatives in Ag-RDTs, a Cochrane review reported variations in sensitivities between brands ranging from 34% to 88% [5]. Even though Ag-RDTs have less sensitivity that real-time polymerase chain reaction (rt-PCR), they reliably identify people with high viral loads, showing high sensitivity that increases with lower cycle threshold (Ct) values (Ct <25, 98.2%; Ct<30, 94.9%) [4]. A previous study demonstrated that Ag-RDTs showed a better correlation with cell culture than rt-PCR [6], so, while rt-PCR is the gold standard for COVID-19 detection, Ag-RDTs are more efficient to detect infectious patients. rt-PCR is a highly sensitive technique that can detect viral RNA for prolonged periods and detects non-transmittable SARS-CoV-2 RNA. This could overestimate the number of contagious patients [7].

Although some contradictory findings have been reported, Ag-RDTs used as self-tests by the general population have a similar accuracy to when they are performed by health professionals [4–7].

Healthcare professionals, who are in contact with many patients, are at a high risk of exposure and, eventually if infected, of transmitting it to vulnerable patients. Above all, the most vulnerable ones such as oncological and immunosuppressed patients. The WHO recommends early detection of SARS-CoV-2 infection among health workers through syndromic surveillance and/or regular testing [8]. Little is known about the level of exposure of teachers and other professionals in the field of education, nevertheless the high transmissibility of the Omicron variant has also dramatically increased prevalence and incidence of SARS-CoV-2 in schools, where exposure is high as well.

Self-testing based on Ag-RDTs could increase access to testing and early confirmation of cases, and thus, reducing transmission. We were interested in investigating the acceptability and feasibility of self-testing in two key populations. We implemented a pilot intervention based on online offering of self-test kits for the SARS-CoV-2 rapid antigen test (TESTA'T COVID) during the peak of the 6[th] wave caused by the Omicron variant of SARS-CoV-2 in Catalonia (Spain). The objective of the study was to assess the feasibility of TESTA'T COVID strategy among healthcare and education professionals.

## Methods

### Study design

Non-randomized prospective study.

### Study sitting and timing

Data were collected prospectively during the peak of the 6[th] wave due to the Omicron variant of SARS-CoV-2 in Catalonia, from 15 December 2021 to 15 February 2022.

### Study population

The study targeted two different key populations: 1) Staff of the Catalan Institute of Oncology (ICO), a public non-profit organization attached to the Catalan Health Service focused on cancer care and with 1,400 professionals distributed in 5 tertiary hospitals in Catalonia (Spain). 2) Staff of the schools belonging to the COVID Sentinel School Network of Catalonia (CSSNC), which monitors SARS-CoV-2 infection and its determinants, by means of repetitive cross-sectional surveys and includes 23 participating sentinel schools and 700 employees [9].

Inclusion criteria were being 18 years old or older, being staff of ICO or CSSNC, and signing the online informed consent.

## Sampling and sample size

Health care workers of the ICO and teachers and school staff from all participating schools of the CSSNC were invited to participate by email. The expected number of participants to recruit was 500. Respondents accessed the study website (https://www.testate.org/), signed up and accepted through an online written and self-complimented informed consent form. Then, participants requested a free COVID-19 rapid lateral flow home test kit (PanBIO ™ COVID-19 Antigen Self-Test, Abbot Laboratories, Chicago, US) and provided contact details including a postal address. Kits included a pictorial leaflet with guidance on how to perform the test and an instructional video was available on YouTube.

## Tools of data collection

After performing the test, participants completed an online and self-complimented survey on the project website including sociodemographic characteristics (gender, age, job position), clinical data (presenting symptoms compatible with COVID-19, days since symptoms" onset, vaccination status, number of vaccine's doses and kind of vaccine), satisfaction with the intervention (Likert-type scale), willingness to repeat the self-test in the future, ease of use of the kit, level of trust in having done a correct interpretation of the obtained result, confidence in the results obtained with Ag self-tests, interest in having available the self-test in their workplace, preferred place to repeat the test (health care centre, do it themselves at home, other), perceived advantages and disadvantages of self-tests, result obtained, and a picture of the result. These pictures were assessed blind by the field coordinator and a microbiologist to check if the reading had been done correctly. The field coordinator and the microbiologist read the results of the pictures uploaded on the project website by the participants without knowing the interpretation made by the participants and by the other member of the research team. All participants with a positive result were contacted and were recommended both to self-isolate and to contact their General Practitioner (GP) as soon as possible.

## Study outcomes data management

The main outcome of the study was to evaluate the feasibility of the TESTATE COVID testing strategy among its users. The assessment was based on a conceptual framework adapted from earlier models [10,11]. The adapted framework divides the concept of feasibility into learnability, willingness, suitability, satisfaction, and efficacy (Fig 1). Learnability was defined as the ability of the participant to understand how to correctly perform the self-test and accurately read the test results. Willingness was defined as the intention of participants to follow all the procedure. Suitability was defined as participants' belief that the test is relevant for their work and that test results are a true indication of the presence or absence of SARS-CoV-2 infection. Satisfaction was described as feeling that being tested for COVID-19 through the TESTATE intervention was convenient and that it is a process they would experience again. Efficacy was defined as participants' ability to make the effort and time to order the self-testing kit, perform the test, report the obtained result, as well as follow the linkage to care procedure if necessary.

The secondary outcomes of the study were the prevalence of SARS-CoV-2 infection among health care and education professionals and the linkage to care rate for those with a positive result. We estimated the SARS-CoV-2 infection prevalence by calculating the proportion of individuals with a positive result over the total number of individuals who uploaded a picture of the obtained. Confidence interval 95% was calculated. The linkage to care rate was assessed by calculating the proportion of individuals who self-reported having contacted their general practitioner and isolated themselves.

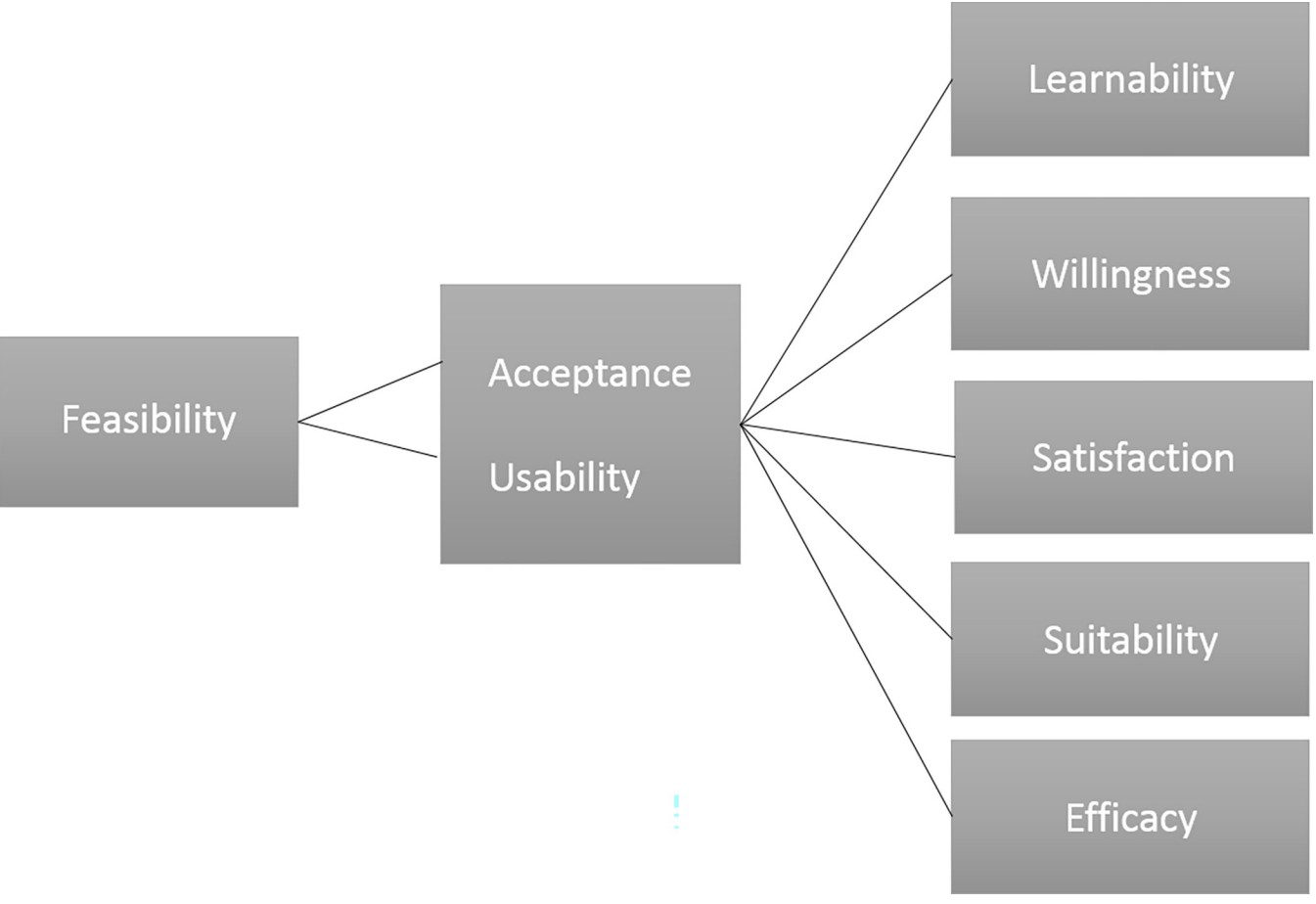

**Fig 1. Conceptual framework for the evaluation of a pilot intervention based on offering online COVID-19 self-test kits addressed to healthcare and education professionals in Spain, adapted from Asiimwe et al. 2012 and Ansbro et al. 2015.**

A descriptive analysis was carried out, comparing socio-demographic and clinical characteristics and acceptance and usability dimensions among health care and education professionals. Qualitative variables were compared using Pearson's $\chi$2 test. Quantitative variable comparisons were made between 2 or more groups using non-parametric tests (Kruskal-Wallis). For all analyses, a significance level of 5% was considered. All analyses were done using R version 4.0.5.

## Ethical considerations

Confidentiality was guaranteed in accordance with the provisions of the Regulation (EU) 2016/679 of the European Parliament and of the Council of 27 April 2016 and the new national Organic Law of Protection of Personal Data (3/2018, 5 December, Data Protection and Digital Rights Act). All participants were provided with online information about the study and given the opportunity to ask questions and clarify queries with the study coordinator by email or phone. The study was approved by the Ethical Committee of the Germans Trias i Pujol Hospital (PI-20-368).

## Results

During the study period 297 educators and 195 health professionals ordered a self-sampling kit, and 192 teachers (64.6%) and 111 health professionals (56.9%) correctly uploaded a picture

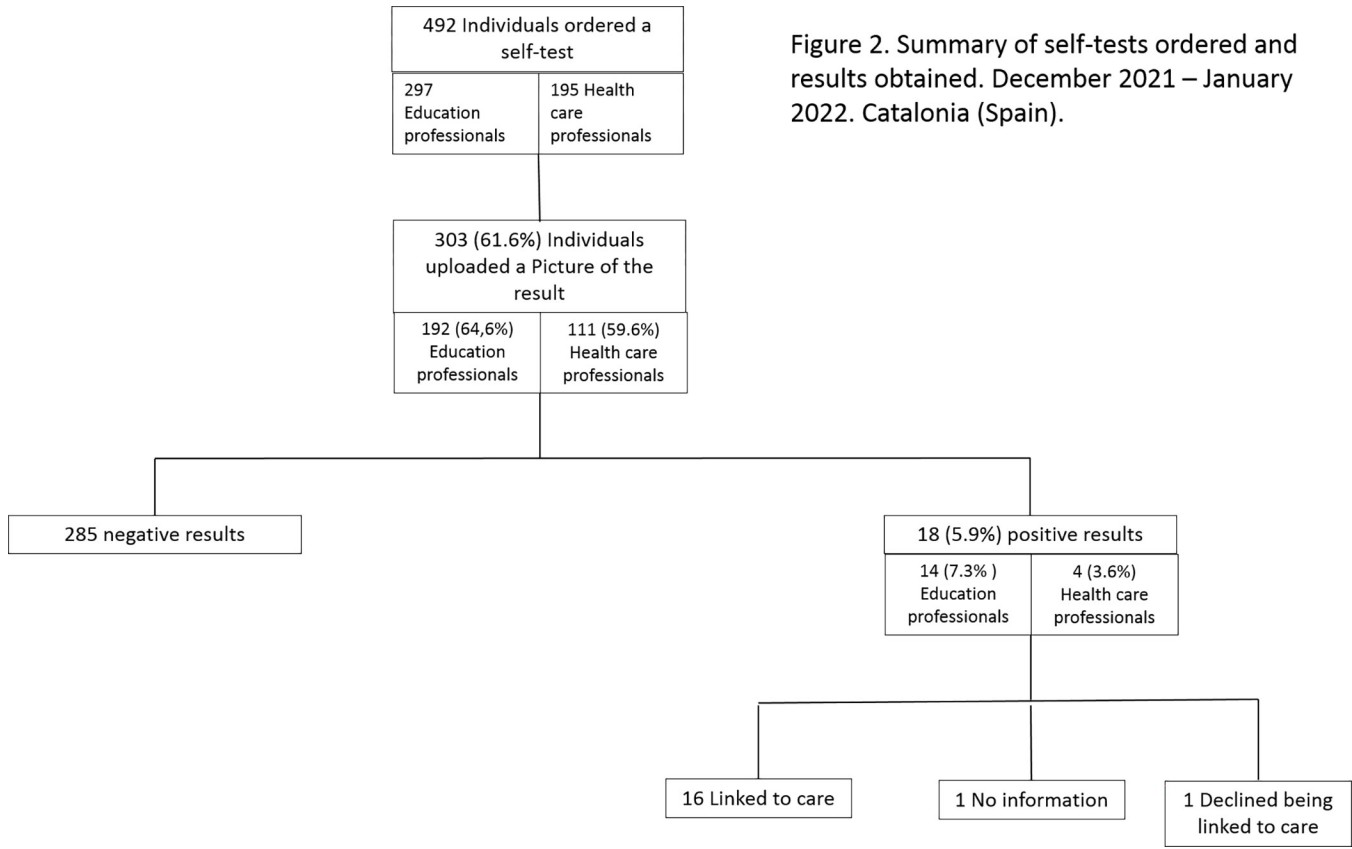

Figure 2. Summary of self-tests ordered and results obtained. December 2021 – January 2022. Catalonia (Spain).

**Fig 2. Summary of self-tests ordered and results obtained.** December 2021 –January 2022. Catalonia (Spain).

of their result and answered the online survey (p: 0.1035) (Fig 2). Sociodemographic and clinical characteristics of participants are shown in Table 1. The median age of participants was 43.0 (IQR: 20.0–78.0), 80.1% were women. Most participants had received the COVID-19 vaccine (98.7%), among those vaccinated, 63.2% had had three doses and 30.0% had had two. No significative differences between health care and education professionals were observed except for the number of received doses of the COVID-19 that was higher for health care professionals because they were one of the first groups who received the third dose. Differences were also observed in the kind of vaccines received, 27% of education professionals were vaccinated with Astra Zeneca (Cambridge, United Kingdom) while none of the health professionals received this vaccine (Table 1).

We detected 18 positive cases, including two cases which were identified as negative by the participants and as positive by the research team (Fig 2). The proportion of positive results was higher among teachers (7.3%) than health care professionals (3.6%) (p: 0.1959). Among positive participants there were: 11 teachers, three physicians, one nurse, two school administrative staff and one person without information.

We estimated a global prevalence of SARS-CoV-2 infection of 5.94% (CI 95%: 3.28, 8.6), being 3.6% (CI 95%: 0.14, 7.07) in health care professionals and 7.29% (CI 95%: 3.61, 10.97) in education professionals (P value: 0.2908).

Most participants (78.5%) with a positive result had symptoms compatible with COVID-19 and all of them but two, contacted their GP and isolated themselves after knowing the result (linkage to care rate: 88.9%). One was not possible to contact and the other one ignored the

**Table 1. Characteristics of the participants in the pilot intervention TESTA'T COVID, Spain December 2021 -January 2022.** N: 307.

| | All n(%) | Education professionals n(%) | Health care professionals n(%) | |
|---|---|---|---|---|
| | N = 307 | N = 196 | N = 111 | p value |
| Gender | | | | 0.694 |
| Women | 246 (80.1%) | 155 (79.1%) | 91 (82.0%) | |
| Men | 59 (19.2%) | 39 (19.9%) | 20 (18.0%) | |
| DK | 2 (0.65%) | 2 (1.02%) | 0 (0.00%) | |
| Age | | | | |
| Median (Interquatilic Range) | 42.4 (10.3) | 42.1 (10.3) | 42.8 (10.4) | 0.587 |
| Country of birth | | | | 0.010 |
| Spain | 294 (95.8%) | 192 (98.0%) | 102 (91.9%) | |
| Other | 12 (3.91%) | 3 (1.53%) | 9 (8.11%) | |
| Dk/Da | 1 (0.33%) | 1 (0.51%) | 0 (0.00%) | |
| Job health care professionals | | | | <0.001 |
| Doctor | 17 (5.54%) | 0 (0.00%) | 17 (15.3%) | |
| Nurse | 29 (9.45%) | 0 (0.00%) | 29 (26.1%) | |
| Nurse assistant | 4 (1.30%) | 0 (0.00%) | 4 (3.60%) | |
| Psychologist | 3 (0.98%) | 0 (0.00%) | 3 (2.70%) | |
| Pharmacist | 4 (1.30%) | 0 (0.00%) | 4 (3.60%) | |
| Administrative staff | 8 (2.61%) | 0 (0.00%) | 8 (7.21%) | |
| Manager | 6 (1.95%) | 0 (0.00%) | 6 (5.41%) | |
| Other | 39 (12.7%) | 0 (0.00%) | 39 (35.1%) | |
| Dk/Da | 1 (0.33%) | 0 (0.00%) | 1 (0.90%) | |
| N/A | 196 (63.8%) | 196 (100%) | 0 (0.00%) | |
| Job education professionals | | | | <0.001 |
| Teacher | 156 (50.8%) | 156 (79.6%) | 0 (0.00%) | |
| Psychologist | 1 (0.33%) | 1 (0.51%) | 0 (0.00%) | |
| Manager | 19 (6.19%) | 19 (9.69%) | 0 (0.00%) | |
| Administrative staff | 6 (1.95%) | 6 (3.06%) | 0 (0.00%) | |
| Responsable/monitor de menjador | 1 (0.33%) | 1 (0.51%) | 0 (0.00%) | |
| Conserge | 3 (0.98%) | 3 (1.53%) | 0 (0.00%) | |
| Altre | 9 (2.93%) | 9 (4.59%) | 0 (0.00%) | |
| Dk/Da | 1 (0.33%) | 1 (0.51%) | 0 (0.00%) | |
| N/A | 111 (36.2%) | 0 (0.00%) | 111 (100%) | |
| Presents covid symptoms | | | | 0.849 |
| Yes | 63 (20.5%) | 42 (21.4%) | 21 (18.9%) | |
| No | 241 (78.5%) | 152 (77.6%) | 89 (80.2%) | |
| Dk/Da | 3 (0.98%) | 2 (1.02%) | 1 (0.90%) | |
| Days since sympthoms' onset | | | | 0.298 |
| Same day | 4 (1.30%) | 3 (1.53%) | 1 (0.90%) | |
| 1 day | 6 (1.95%) | 3 (1.53%) | 3 (2.70%) | |
| 2 days | 14 (4.56%) | 11 (5.61%) | 3 (2.70%) | |
| 3 days | 7 (2.28%) | 5 (2.55%) | 2 (1.80%) | |
| 4 days | 9 (2.93%) | 4 (2.04%) | 5 (4.50%) | |
| 5 days | 7 (2.28%) | 6 (3.06%) | 1 (0.90%) | |
| 6 days | 1 (0.33%) | 1 (0.51%) | 0 (0.00%) | |
| 7 days | 4 (1.30%) | 4 (2.04%) | 0 (0.00%) | |
| > 7 days | 10 (3.26%) | 5 (2.55%) | 5 (4.50%) | |

*(Continued)*

**Table 1.** (Continued)

| | | Education professionals | Health care professionals | |
|---|---|---|---|---|
| | All n(%) | n(%) | n(%) | |
| | N = 307 | N = 196 | N = 111 | p value |
| Dk/Da | 2 (0.65%) | 0 (0.00%) | 2 (1.80%) | |
| N/A | 243 (79.2%) | 154 (78.6%) | 89 (80.2%) | |
| Has been vaccinated for COVID-19 | | | | 1.000 |
| Si | 303 (98.7%) | 193 (98.5%) | 110 (99.1%) | |
| No | 3 (0.98%) | 2 (1.02%) | 1 (0.90%) | |
| Dk/Da | 1 (0.33%) | 1 (0.51%) | 0 (0.00%) | |
| Number of vaccine's doses | | | | <0.001 |
| One | 18 (5.86%) | 17 (8.67%) | 1 (0.90%) | |
| Two | 92 (30.0%) | 75 (38.3%) | 17 (15.3%) | |
| Three | 194 (63.2%) | 101 (51.5%) | 93 (83.8%) | |
| N/A | 3 (0.98%) | 3 (1.53%) | 0 (0.00%) | |
| Type of vaccine | | | | <0.001 |
| Pfizer | 69 (22.5%) | 44 (22.4%) | 25 (22.5%) | |
| Moderna | 171 (55.7%) | 91 (46.4%) | 80 (72.1%) | |
| Astra Zeneca | 53 (17.3%) | 53 (27.0%) | 0 (0.00%) | |
| Jansen | 1 (0.33%) | 1 (0.51%) | 0 (0.00%) | |
| Other | 8 (2.61%) | 4 (2.04%) | 4 (3.60%) | |
| Dk/Da | 1 (0.33%) | 0 (0.00%) | 1 (0.90%) | |
| N/A | 4 (1.30%) | 3 (1.53%) | 1 (0.90%) | |

recommendations because she considered herself to be negative, although the research team read the result as positive and informed her of it.

## Learnability

The majority of the participants (92.5%) found that the self-test was easy or very easy to use 99.7% successfully completed the test and, 88.9% did not need any help to perform the test no significant differences were observed between healthcare and education professionals. 93.5% agreed or strongly agreed with the statement "I trust that my interpretation of the result I obtained with the self-test is correct", 81.6% of educators strongly agreed compared to 66.7% health professionals (P: 0.006) (Table 1). Two (0,6%) participants failed to read the test results.

## Willingness

Most of participants (96.7%) agreed or strongly agreed with the statement "I would repeat the rapid SARS-CoV-2 antigen self-test in the future", 90.3% of educators strongly agreed compared with 80.2% of health professionals (p: 0.021). The most preferred way to repeat the test was "do the self-test at home" (87.9%); and 99.3% would like the test to be available at their workplace, with no differences among health and education professionals (Table 2).

## Suitability

Most participants (89.6%) agreed or strongly agreed with the statement "I trust the result obtained with the self-test" (Table 2).

**Table 2.** Learability, willingness, suitability and satisfaction of the TESTATE COVID intervention adressed to health care and education professionals in Catalonia (Spain), N: 307. December 2021-February 2022.

| | All n(%) | Education professionals n(%) | Health care professionals n(%) | |
|---|---|---|---|---|
| **Learnability[1]** | N = 307 | N = 196 | N = 111 | p value |
| Test difficulty | | | | 0.479 |
| Very easy | 219 (71.3%) | 145 (74.0%) | 74 (66.7%) | |
| Easy | 65 (21.2%) | 38 (19.4%) | 27 (24.3%) | |
| Neither easy or difficult | 17 (5.5%) | 10 (5.1%) | 7 (6.3%) | |
| Difficult | 5 (1.6%) | 2 (1.0%) | 3 (2.7%) | |
| Very difficult | 0 (0.0%) | 0 (0.0%) | 0 (0.0%) | |
| Dk/Da | 1 (0.3%) | 1 (0.5%) | 0 (0.0%) | |
| Have successfully completed the self-test | | | | 1.000 |
| No | 0 (0.0%) | 0 (0.0%) | 0 (0.0%) | |
| Yes | 306 (99.7%) | 195 (99.5%) | 111 (100%) | |
| Dk/Da | 1 (0.3%) | 1 (0.5%) | 0 (0.0%) | |
| Needed help to perform the self-test | | | | 0.523 |
| No | 273 (88.9%) | 171 (87.2%) | 102 (91.9%) | |
| Yes | 33 (10.7%) | 24 (12.2%) | 9 (8.1%) | |
| Dk/Da | 1 (0.3%) | 1 (0.5%) | 0 (0.0%) | |
| Trusting that the interpretation of the result obtained with the self-test is correct | | | | 0.006 |
| Strongly agree | 234 (76.2%) | 160 (81.6%) | 74 (66.7%) | |
| Agree | 53 (17.3%) | 24 (12.2%) | 29 (26.1%) | |
| Neither agree nor disagree | 9 (2.9%) | 6 (3.1%) | 3 (2.7%) | |
| Disagree | 7 (2.3%) | 3 (1.5%) | 4 (3.6%) | |
| Strongly disagree | 1 (0.3%) | 0 (0.0%) | 1 (0.9%) | |
| Dk/Da | 3 (1.0%) | 3 (1.5%) | 0 (0.0%) | |
| **Willingness[2]** | | | | |
| Would repeat the self-test in the future | | | | 0.021 |
| Strongly agree | 266 (86.6%) | 177 (90.3%) | 89 (80.2%) | |
| Agree | 31 (10.1%) | 16 (8.2%) | 15 (13.5%) | |
| Neither agree nor disagree | 3 (1.0%) | 1 (0.5%) | 2 (1.8%) | |
| Disagree | 3 (1.0%) | 0 (0.0%) | 3 (2.7%) | |
| Strongly disagree | 1 (0.3%) | 0 (0.0%) | 1 (0.9%) | |
| Dk/Da | 3 (1.0%) | 2 (1.0) | 1 (0.9%) | |
| Preferred place to repeat the self-tets in the future | | | | 0.241 |
| Health care centre | 22 (7.2%) | 18 (9.2%) | 4 (3.6%) | |
| Do the self-test at home | 270 (87.9%) | 170 (86.7%) | 100 (90.1%) | |
| Other | 6 (1.9%) | 3 (1.5%) | 3 (2.7%) | |
| Dk/Da | 9 (2.6%) | 5 (2.5%) | 4 (3.6%) | |
| Would like to have available self-tests at their workplace | | | | 0.595 |
| Yes | 304 (99.3%) | 194 (99.5%) | 110 (99.1%) | |
| No | 1 (0.33%) | 1 (0.5%) | 0 (0.0%) | |
| Dk/Da | 1 (0.33%) | 0 (0.0%) | 1 (0.9%) | |

*(Continued)*

**Table 2.** (Continued)

| | All n(%) | Education professionals n(%) | Health care professionals n(%) | |
|---|---|---|---|---|
| **Learnability**[1] | N = 307 | N = 196 | N = 111 | p value |
| **Suitability**[3] | | | | |
| Confidence in the results obtained with the SARS-CoV-2 rapid antigen self-tests | | | | 0.236 |
| Strongly agree | 158 (51.5%) | 105 (53.6%) | 53 (47.7%) | |
| Agree | 117 (38.1%) | 69 (35.2%) | 48 (43.2%) | |
| Neither agree nor disagree | 18 (5.9%) | 14 (7.1%) | 4 (3.6%) | |
| Disagree | 10 (3.3%) | 6 (3.1%) | 4 (3.6%) | |
| Strongly disagree | 2 (0.6%) | 0 (0.0%) | 2 (1.8%) | |
| Dk/Da | 2 (0.6%) | 2 (1.0%) | 0 (0.0%) | |
| **Satisfaction**[4] | | | | |
| Test satisfaction | | | | 0.314 |
| Very satisfied | 236 (76.9%) | 153 (78.1%) | 83 (74.8%) | |
| Satisfied | 47 (15.3%) | 26 (13.3%) | 21 (18.9%) | |
| Neither satisfied or unsatisfied | 20 (6.51%) | 15 (7.65%) | 5 (4.50%) | |
| Unsatisfied | 3 (0.98%) | 1 (0.51%) | 2 (1.80%) | |
| Very unsatisfied | 0 (0.00%) | 0 (0.00%) | 0 (0.00%) | |
| Dk/Da | 1 (0.33%) | 1 (0.51%) | 0 (0.00%) | |
| Would recommend the self-test to a friend | | | | 0.024 |
| Strongly agree | 246 (80.1%) | 160 (81.6%) | 86 (77.5%) | |
| Agree | 42 (13.7%) | 24 (12.2%) | 18 (16.2%) | |
| Neither agree nor disagree | 8 (2.6%) | 8 (4.1%) | 0 (0.0%) | |
| Disagree | 8 (2.6%) | 3 (1.5%) | 5 (4.5%) | |
| Strongly disagree | 1 (0.3%) | 0 (0.0%) | 1 (0.9%) | |
| Dk/Da | 2 (0.7%) | 1 (0.5%) | 1 (0.9%) | |
| Advantges of the self-test | | | | |
| Self-test can improve security and protection of COVID-19 in the workplace | 260 (84.7%) | 170 (86.7%) | 90 (81.1%) | 0.247 |
| Obtention of the results in a few minutes | 262 (85.3%) | 163 (83.2%) | 99 (89.2%) | 0.205 |
| Privacy and confidentiality | 111 (36.2%) | 62 (31.6%) | 49 (44.1%) | 0.039 |
| Convenience | 258 (84.0%) | 167 (85.2%) | 91 (82.0%) | 0.563 |
| Test is free | 195 (63.5%) | 127 (64.8%) | 68 (61.3%) | 0.621 |
| Do not need naso-pharyngeal swab | 51 (16.6%) | 28 (14.3%) | 23 (20.7%) | 0.195 |
| Do not need to explain yourself to others | 41 (13.4%) | 22 (11.2%) | 19 (17.1%) | 0.199 |
| Self-test contributes to normalize COVID-10 testing | 131 (42.7%) | 82 (41.8%) | 49 (44.1%) | 0.785 |
| Self-tests allow taking control of our health | 164 (53.4%) | 108 (55.1%) | 56 (50.5%) | 0.505 |
| Self-tests give more sense of security at the work place | 168 (54.7%) | 122 (62.2%) | 46 (41.4%) | 0.001 |
| Other | 3 (1.0%) | 2 (1.0%) | 1 (0.9%) | 1.000 |
| Disadvantges of the self-test | | | | |

(*Continued*)

**Table 2.** (Continued)

| | All n(%) | Education professionals n(%) | Health care professionals n(%) | |
|---|---|---|---|---|
| **Learnability[1]** | **N = 307** | **N = 196** | **N = 111** | **p value** |
| None | 275 (89.6%) | 173 (88.3%) | 102 (91.9%) | 0.421 |
| Sensitivity and specificity lower than a PCR | 17 (5.5%) | 10 (5.1%) | 7 (6.3%) | 0.854 |
| Having to interpretate the result by yourself | 12 (3.9%) | 6 (3.1%) | 6 (5.4%) | 0.363 |
| The time for obtaining the result is too long | 1 (0.3%) | 0 (0.0%) | 1 (0.9%) | 0.362 |
| Other | 21 (6.8%) | 10 (5.1%) | 11 (9.9%) | 0.171 |
| Don't know | 16 (5.2%) | 13 (6.6%) | 3 (2.7%) | 0.222 |
| Don't want to answer | 3 (1.0%) | 3 (1.5%) | 0 (0.0%) | 0.241 |

1. **Learnability** was defined as the ability of the participant to understand how to correctly perform the self-test and accurately read the test results.

2. **Willingness** was defined as the intention of participants to follow all the procedure.

3. **Suitability** was defined as participants' belief that the test is relevant for their work and that test results are a true indication of the presence or absence of SARS-CoV-2 infection.

4. **Satisfaction** was described as feeling that being tested for COVID-19 through the TESTATE intervention was convenient and that it is a process they would experience again. Efficacy was defined as participants' ability to make the effort and time to order the self-testing kit, perform the test, report the obtained result, as well as follow the linkage to care procedure if necessary.

## Satisfaction

92.2% of the participants answered that they were satisfied or very satisfied with the intervention; and 93.8% agreed or strongly agreed with the statement "I would recommend it to a friend" with 81.6% of educators strongly agreeing compared to 77.5% of health professionals (p:0.024). The advantages that were most identified were getting the result in a few minutes (85.3%) and the fact that the tests might improve safety and protection against COVID-19 at their workplace (84.7%); educators were more likely to identify "Self-tests give more sense of security at the workplace" as an advantage than health professionals (62.2% vs. 41.4%, p: 0.001); 89.6% did not identify any disadvantages.

## Discussion

Testing is a critical component of the overall prevention and control strategy for the COVID-19 pandemic [12]. Nevertheless, apart from contact tracing strategies, screening of key populations implies many logistical and operational challenges, including the necessity of periodic testing in periods of high incidence (ex. twice a week) in order to be effective [2]. Highly sensitive self-tests are cheap, simple, rapid tests and that can enable high frequency regimens that will capture most infections while they are still infectious.

Uploading the pictures of the results online contributed to better traceability of positives and this could improve the ability to break the epidemiological chain.

Although the study used an opportunistic sample that is not representative of the healthcare and education professionals of the region, we demonstrated that the TESTA'T COVID intervention is feasible. We provided an in-depth account of acceptability and usability of an online screening strategy based on antigen self-tests for COVID-19 addressed to health care and education professionals in Catalonia. Our study showed high feasibility of the intervention both in healthcare and education professionals, although education professionals presented higher learnability with higher level of trust in having a correct interpretation of the obtained results; and, higher willingness to repeat the self-test in the future and to recommend it to a friend. In the event of a possible consolidation of the pilot intervention, the implementation of

campaigns to increase the level of trust and acceptability of self-tests by healthcare professionals should be considered.

High proportion of participants tested positive (88.9%) were able to correctly read their result, similarly to previous studies [13]. Errors might be reduced by refinement of the instructions provided.

Apart from showing a high feasibility our pilot intervention showed high efficacy in terms of number of tests requested, number of correctly uploaded picture of the result, answered surveys, and positivity and linkage to care rates). Linkage to care is challenging in self-sampling strategies, however we obtained high percentages of linkage to care (88.9%).

This is the first time this was done in Spain, and during the peak of the $6^{th}$ wave caused by the Omicron variant. Our pilot intervention has been proved feasible and has the potential for frequent and extensive testing. The generated information on acceptability and usability, as well as positivity and linkage to acre rates, will be crucial to better define tailored screening strategies addressed to specific key populations, particularly during peaks of high community transmission of SARS-CoV-2 and eventually other respiratory transmitted agents.

## Supporting information

**S1 Data.**
(XLSX)

**S2 Data.**
(XLSX)

## Acknowledgments

The authors acknowledge the collaboration of Andreu Colom, Gema Ballega, Juan Rus, Marina Herrero, Pili Bonamusa, the General Direction of the Health Department and the Education Department of the Catalan Government of Catalonia, Anna Clopés, Deputy Director of the Catalan Institute of Oncology (ICO), Ana Sedano, Director of Human Resources of ICO and Abbot Laboratories.

## Author Contributions

**Conceptualization:** Cristina Agustí, Jordi Casabona.

**Formal analysis:** Yesika Díaz, Marcos Montoro-Fernandez, Sergio Moreno-Fornés.

**Funding acquisition:** Jordi Casabona.

**Investigation:** Cristina Agustí, Héctor Martínez-Riveros, Victoria González, Gema Fernández-Rivas, Yesika Díaz, Marcos Montoro-Fernandez, Sergio Moreno-Fornés, Pol Romano-deGea, Esteve Muntada, Beatriz Calvo, Jordi Casabona.

**Methodology:** Victoria González, Gema Fernández-Rivas.

**Software:** Esteve Muntada.

**Supervision:** Cristina Agustí, Jordi Casabona.

**Visualization:** Pol Romano-deGea.

**Writing – original draft:** Cristina Agustí.

**Writing – review & editing:** Héctor Martínez-Riveros, Victoria González, Gema Fernández-Rivas, Yesika Díaz, Marcos Montoro-Fernandez, Sergio Moreno-Fornés, Pol Romano-deGea, Esteve Muntada, Beatriz Calvo, Jordi Casabona.

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
