## [Decision Letter · Decision Letter 0]

26 May 2022

PONE-D-22-12378Feasibility of an online antigen self-testing strategy for SARS-CoV-2 addressed to health care and education professionals in Catalonia (Spain). The TESTA’T- COVID Project PLOS ONE

Dear Dr. Agustí,

Thank you for submitting your manuscript to PLOS ONE. After careful consideration, we feel that it has merit but does not fully meet PLOS ONE’s publication criteria as it currently stands. Therefore, we invite you to submit a revised version of the manuscript that addresses the points raised during the review process.

ACADEMIC EDITOR:

Respected authors your manuscript is good. However; your research manuscript needs great attention from your side to reach to required level for publishing specially regarding; the objectives for this study which need to be clear and specific. The method section which needs to be rigorously organized in a proper scientific way. The results ; be sure that your table is inserted within the manuscript and not in the attached other files.

  <g-bubble data-du="200" data-tp="5" jsaction="R9S7w:VqIRre;" jscontroller="QVaUhf" jsshadow="">

<g-img aria-label="رمز "تم التحقق منها بواسطة المنتدى"" class="XrZwB" role="img" style="display: inline-block; height: 26px; padding-right: 5px; vertical-align: middle;"></g-img></g-bubble>  

We look forward to receiving your revised manuscript.

Kind regards,

Omnia Samir El Seifi, Professor Ph.D

Academic Editor

PLOS ONE

Journal Requirements:

"I have read the journal's policy and the authors of this manuscript have the following competing interests: Abbot Laboratories provided the tests for the study for free"

5. Please include your tables as part of your main manuscript and remove the individual files. Please note that supplementary tables (should remain/ be uploaded) as separate "supporting information" files.

Reviewers' comments:

Reviewer's Responses to Questions

**Comments to the Author**

1. Is the manuscript technically sound, and do the data support the conclusions?

Reviewer #1: Partly

Reviewer #2: Partly

2. Has the statistical analysis been performed appropriately and rigorously? 

Reviewer #1: I Don't Know

Reviewer #2: I Don't Know

3. Have the authors made all data underlying the findings in their manuscript fully available?

Reviewer #1: Yes

Reviewer #2: No

4. Is the manuscript presented in an intelligible fashion and written in standard English?

Reviewer #1: No

Reviewer #2: Yes

5. Review Comments to the Author

Reviewer #1: the respected authors presented a good idea that would add to the available methods of screening. But it is poorly presented , many changes are needed to be suitable for publication.

abstract: the aim is not properly formulated. changes will be needed based on the next points.

introduction: what is the knowledge gap that you are trying to cross-over.

the rationale is missed.

the aim of the study is vague and not fulfilled in the manuscript. you may change it to be:"to assess the feasibility of the TESTATE COVID screening strategy among.......".

Methods:

Add a figure to illustrate the flow of recruitment process.

move the illustration/definition of the framework components to the methods section.

what are your study outcomes?

The discussion is very confusing, the term "screening" is not properly used. the authors need to clarify what is their target. you may recommend using the kit based on your findings, but in the current status I can't get what is your target.

you need to discuss the main findings, the difference between your 2 groups, and so on ...

The conclusion and recommendation are deficient.

Reviewer #2: The authors have done a good job/

Here are my comments:

Introduction:

- More data from the literature should be mentioned to show the complexity of detecting COVID cases, the use of the antigen tests, and their limitations.

- The references used are very limited, more evidence should be added to support the study's claim.

Methodology:

- The interpretation of test results by the professional team is not clear.

- Nothing was mentioned regarding the statistical analysis of data.

- More data regarding inclusion criteria and procedures for collecting the survey data are needed.

- The used framework for evaluation is not well explained.

Results:

- Adding one or two figures illustrating the data mentioned, especially those from the survey would benefit the presentation of data

- The results should be statistically analyzed and presented in the results section

Discussion:

- The discussion section should include more previous studies that are comparable or related to the current study.

- The part of the evaluation should be explained more with relevance to the above-mentioned framework to be able to give a conclusion on the overall feasibility.

Conclusion:

More conclusions of the study should be added at the end

6. PLOS authors have the option to publish the peer review history of their article (what does this mean?). If published, this will include your full peer review and any attached files.

Reviewer #1: **Yes: **rehab Hosny El-Sokkary

Reviewer #2: No

---

## [Author Response · Author response to Decision Letter 0]

28 Jul 2022

PONE-D-22-12378

Feasibility of an online antigen self-testing strategy for SARS-CoV-2 addressed to health care and education professionals in Catalonia (Spain). The TESTA’T- COVID Project

COMMENTS TO THE AUTHORS

ACADEMIC EDITOR:

Respected authors your manuscript is good. However; your research manuscript needs great attention from your side to reach to required level for publishing specially regarding; the objectives for this study which need to be clear and specific. 

Answer: Following the recommendations of the Editor and the reviewers, the objectives have been re-written.

The method section has been completed and re-organized in a proper scientific way. 

Answer: 

The results; be sure that your table is inserted within the manuscript and not in the attached other files.

Answer: The tables have been inserted within the manuscript.

Journal Requirements:

Answer:

Answer: Additional details regarding the informed consent have been included in the new version of the manuscript.

"I have read the journal's policy and the authors of this manuscript have the following competing interests: Abbot Laboratories provided the tests for the study for free"

Answer: The updated Competing Interests statement has been included in the Competing Interests section and in the cover letter.

Answer: In the informed consent approved by the Ethical Committee of the Germans Trias i Pujol Hospital (PI-20-368) it was included the following statement: No sharing data with third party is planned. For this reason, we understand we are not able to share data of the included participants. 

5. Please include your tables as part of your main manuscript and remove the individual files. Please note that supplementary tables (should remain/ be uploaded) as separate "supporting information" files.

Answer: Done.

Reviewer's Responses to Questions

Comments to the Author

1. Is the manuscript technically sound, and do the data support the conclusions?

Reviewer #1: Partly

Reviewer #2: Partly

Answer: We re-wrote the manuscript to solve this issue.

2. Has the statistical analysis been performed appropriately and rigorously?

Reviewer #1: I Don't Know

Reviewer #2: I Don't Know

Answer: We further explained the statistical data analysis performed in the new version of the manuscript.

3. Have the authors made all data underlying the findings in their manuscript fully available?

Reviewer #1: Yes

Reviewer #2: No

Answer: In the informed consent approved by the Ethical Committee of the Germans Trias i Pujol Hospital (PI-20-368) it was included the following statement: No sharing data with third party is planned. For this reason, we understand we are not able to share data of the included participants. 

4. Is the manuscript presented in an intelligible fashion and written in standard English?

 Reviewer #1: No

Reviewer #2: Yes

Answer: We re-wrote the manuscript to solve this issue.

5. Review Comments to the Author

Reviewer #1: the respected authors presented a good idea that would add to the available methods of screening. But it is poorly presented, many changes are needed to be suitable for publication.

Abstract: the aim is not properly formulated. Changes will be needed based on the next points.

Answer: Following the recommendation of the reviewer, the aim has been reformulated in the abstract and in the introduction section.

Introduction: what is the knowledge gap that you are trying to cross-over.

Answer: The introduction section has been extended.

The rationale is missed.

Answer: The rationale has been included in the introduction section.

The aim of the study is vague and not fulfilled in the manuscript. You may change it to be:"to assess the feasibility of the TESTATE COVID screening strategy among.......".

Answer: Following the recommendation of the reviewer, the aim has been reformulated in the abstract and in the introduction section.

Methods:

Add a figure to illustrate the flow of recruitment process.

Answer: Done. See Figure 2.

Move the illustration/definition of the framework components to the methods section.

Answer: Done. A figure of the framework components has been included in the methods section and the definition of each component has been added in the text.

What are your study outcomes?

Answer: An explanation of what are the study outcomes has been included in the methods section.

The discussion is very confusing, the term "screening" is not properly used. the authors need to clarify what is their target. you may recommend using the kit based on your findings, but in the current status I can't get what is your target.

you need to discuss the main findings, the difference between your 2 groups, and so on ...

Answer: We clarified and extended the Discussion section. We agreed with the reviewer that it was not clear enough. We also agreed that the term screening was not properly used and we correct this issue. 

The conclusion and recommendation are deficient.

Answer: We included more conclusions and recommendations in the Discussion section. 

Reviewer #2: The authors have done a good job/

Here are my comments:

Introduction:

- More data from the literature should be mentioned to show the complexity of detecting COVID cases, the use of the antigen tests, and their limitations.

Answer: The introduction section has been extended to add more information from the literature to show the complexity of detecting COVID cases, the use of the antigen tests, and their limitations.

- The references used are very limited, more evidence should be added to support the study's claim.

Answer: The introduction section has been extended to add more evidence.

Methodology:

- The interpretation of test results by the professional team is not clear.

Answer: We added an explanation about the procedure in the Methods section.

- Nothing was mentioned regarding the statistical analysis of data.

Answer: An explanation of the statistical analysis of data has been added in the Methodology section of the new version of the manuscript.

- More data regarding inclusion criteria and procedures for collecting the survey data are needed.

Answer: An explanation of the inclusion criteria has been added in the Methodology section of the new version of the manuscript

- The used framework for evaluation is not well explained.

Answer: An explanation of the framework used for the evaluation has been added in the Methodology section of the new version of the manuscript

Results:

- Adding one or two figures illustrating the data mentioned, especially those from the survey would benefit the presentation of data

Answer: An extra table and two figures have been included in the new version of the manuscript.

- The results should be statistically analyzed and presented in the results section

Answer: The results section has been modified. We added the p value and the interval of confidence of data presented.

Discussion:

- The discussion section should include more previous studies that are comparable or related to the current study.

Answer: Little information is available on pilot intervention that offer self-tests to detect SARS-CoV-2 infection. Some studies have been included in the introduction and the discussion.

- The part of the evaluation should be explained more with relevance to the above-mentioned framework to be able to give a conclusion on the overall feasibility.

Answer: The part of the evaluation has been explained more and we included an overall conclusion of the feasibility.

Conclusion:

More conclusions of the study should be added at the end.

Answer: More conclusions have been included in the Discussion section.

6. PLOS authors have the option to publish the peer review history of their article (what does this mean?). If published, this will include your full peer review and any attached files.

Do you want your identity to be public for this peer review? For information about this choice, including consent withdrawal, please see our Privacy Policy.

Reviewer #1: Yes: rehab Hosny El-Sokkary

Reviewer #2: No

---

## [Decision Letter · Decision Letter 1]

29 Aug 2022

PONE-D-22-12378R1Feasibility of an online antigen self-testing strategy for SARS-CoV-2 addressed to health care and education professionals in Catalonia (Spain). The TESTA’T- COVID ProjectPLOS ONE

Dear Dr. Agustí,

Thank you for submitting your manuscript to PLOS ONE. After careful consideration, we feel that it has merit but does not fully meet PLOS ONE’s publication criteria as it currently stands. Therefore, we invite you to submit a revised version of the manuscript that addresses the points raised during the review process.

ACADEMIC EDITOR:Dear authors you did a great job, only few changes need to bee done- the objective in the abstract must be as at the end of the introduction- in the method section ; it needs to be organized under subheadings  to make your work more clear and please write them in order as; study design, study sitting and timing, study population, sampling and sample size, tools of data collection, study outcomes data mangement, ethical consideration.

We look forward to receiving your revised manuscript.

Kind regards,

Omnia Samir El Seifi, Professor

Academic Editor

PLOS ONE

Journal Requirements:

Reviewers' comments:

Reviewer's Responses to Questions

**Comments to the Author**

1. If the authors have adequately addressed your comments raised in a previous round of review and you feel that this manuscript is now acceptable for publication, you may indicate that here to bypass the “Comments to the Author” section, enter your conflict of interest statement in the “Confidential to Editor” section, and submit your "Accept" recommendation.

Reviewer #1: (No Response)

2. Is the manuscript technically sound, and do the data support the conclusions?

Reviewer #1: (No Response)

3. Has the statistical analysis been performed appropriately and rigorously? 

Reviewer #1: (No Response)

4. Have the authors made all data underlying the findings in their manuscript fully available?

Reviewer #1: (No Response)

5. Is the manuscript presented in an intelligible fashion and written in standard English?

Reviewer #1: (No Response)

6. Review Comments to the Author

Reviewer #1: Thanks for addressing all raised concern

only one is left: write the aim in the abstract section like that mentioned in the text.

7. PLOS authors have the option to publish the peer review history of their article (what does this mean?). If published, this will include your full peer review and any attached files.

Reviewer #1: **Yes: **rehab Hosny El-Sokkary

---

## [Author Response · Author response to Decision Letter 1]

8 Sep 2022

PONE-D-22-12378R1

Feasibility of an online antigen self-testing strategy for SARS-CoV-2 addressed to health care and education professionals in Catalonia (Spain). The TESTA’T- COVID Project

PLOS ONE

Dear Dr. Agustí,

Thank you for submitting your manuscript to PLOS ONE. After careful consideration, we feel that it has merit but does not fully meet PLOS ONE’s publication criteria as it currently stands. Therefore, we invite you to submit a revised version of the manuscript that addresses the points raised during the review process.

ACADEMIC EDITOR:

Dear authors you did a great job, only few changes need to bee done

- the objective in the abstract must be as at the end of the introduction

- in the method section ; it needs to be organized under subheadings to make your work more clear and please write them in order as; study design, study sitting and timing, study population, sampling and sample size, tools of data collection, study outcomes data mangement, ethical consideration.

Answer: The authors have included the suggested changes in the reviewed version of the manuscript.

• A rebuttal letter that responds to each point raised by the academic editor and reviewer(s). You should upload this letter as a separate file labeled 'Response to Reviewers'. Done.

• A marked-up copy of your manuscript that highlights changes made to the original version. You should upload this as a separate file labeled 'Revised Manuscript with Track Changes'. Done

• An unmarked version of your revised paper without tracked changes. You should upload this as a separate file labeled 'Manuscript'. Done

Not necessary.

We look forward to receiving your revised manuscript.

Kind regards,

Omnia Samir El Seifi, Professor

Academic Editor

PLOS ONE

Journal Requirements:

Please review your reference list to ensure that it is complete and correct. If you have cited papers that have been retracted, please include the rationale for doing so in the manuscript text, or remove these references and replace them with relevant current references. Any changes to the reference list should be mentioned in the rebuttal letter that accompanies your revised manuscript. If you need to cite a retracted article, indicate the article’s retracted status in the References list and also include a citation and full reference for the retraction notice. Done

Reviewers' comments:

Reviewer's Responses to Questions

Comments to the Author

1. If the authors have adequately addressed your comments raised in a previous round of review and you feel that this manuscript is now acceptable for publication, you may indicate that here to bypass the “Comments to the Author” section, enter your conflict of interest statement in the “Confidential to Editor” section, and submit your "Accept" recommendation.

Reviewer #1: (No Response)

2. Is the manuscript technically sound, and do the data support the conclusions?

Reviewer #1: (No Response)

3. Has the statistical analysis been performed appropriately and rigorously? 

Reviewer #1: (No Response)

4. Have the authors made all data underlying the findings in their manuscript fully available?

Reviewer #1: (No Response)

5. Is the manuscript presented in an intelligible fashion and written in standard English?

Reviewer #1: (No Response)

6. Review Comments to the Author

Reviewer #1: Thanks for addressing all raised concern

only one is left: write the aim in the abstract section like that mentioned in the text.

Answer: We have included the aim in the abstract section like that mentioned in the text.

7. PLOS authors have the option to publish the peer review history of their article (what does this mean?). If published, this will include your full peer review and any attached files.

Do you want your identity to be public for this peer review? For information about this choice, including consent withdrawal, please see our Privacy Policy.

Reviewer #1: Yes: rehab Hosny El-Sokkary

---

## [Editor Report · Decision Letter 2]

9 Sep 2022

Feasibility of an online antigen self-testing strategy for SARS-CoV-2 addressed to health care and education professionals in Catalonia (Spain). The TESTA’T- COVID Project

PONE-D-22-12378R2

Dear Dr. Agustí,

We’re pleased to inform you that your manuscript has been judged scientifically suitable for publication and will be formally accepted for publication once it meets all outstanding technical requirements.

Kind regards,

Omnia Samir El Seifi, Professor

Academic Editor

PLOS ONE
---

## [Editor Report · Acceptance letter]

14 Sep 2022

PONE-D-22-12378R2 

Feasibility of an online antigen self-testing strategy for SARS-CoV-2 addressed to health care and education professionals in Catalonia (Spain). The TESTA’T- COVID Project. 

Dear Dr. Agustí:

I'm pleased to inform you that your manuscript has been deemed suitable for publication in PLOS ONE. Congratulations! Your manuscript is now with our production department. 

Kind regards, 

on behalf of

Professor Omnia Samir El Seifi 

Academic Editor

PLOS ONE